# CO_2_-Sensitive Connexin Hemichannels in Neurons and Glia: Three Different Modes of Signalling?

**DOI:** 10.3390/ijms22147254

**Published:** 2021-07-06

**Authors:** Emily Hill, Nicholas Dale, Mark J. Wall

**Affiliations:** School of Life Sciences, University of Warwick, Coventry CV4 7AL, UK; n.e.dale@warwick.ac.uk

**Keywords:** connexin, hemichannel, neuron, glial cell, carbon dioxide, substantia nigra, ventral tegmental area, development, excitability, neuronal signalling

## Abstract

Connexins can assemble into either gap junctions (between two cells) or hemichannels (from one cell to the extracellular space) and mediate cell-to-cell signalling. A subset of connexins (Cx26, Cx30, Cx32) are directly sensitive to CO_2_ and fluctuations in the level within a physiological range affect their open probability, and thus, change cell conductance. These connexins are primarily found on astrocytes or oligodendrocytes, where increased CO_2_ leads to ATP release, which acts on P2X and P2Y receptors of neighbouring neurons and changes excitability. CO_2_-sensitive hemichannels are also found on developing cortical neurons, where they play a role in producing spontaneous neuronal activity. It is plausible that the transient opening of hemichannels allows cation influx, leading to depolarisation. Recently, we have shown that dopaminergic neurons in the substantia nigra and GABAergic neurons in the VTA also express Cx26 hemichannels. An increase in the level of CO_2_ results in hemichannel opening, increasing whole-cell conductance, and decreasing neuronal excitability. We found that the expression of Cx26 in the dopaminergic neurons in the substantia nigra at P7-10 is transferred to glial cells by P17-21, displaying a shift from being inhibitory (to neuronal activity) in young mice, to potentially excitatory (via ATP release). Thus, Cx26 hemichannels could have three modes of signalling (release of ATP, excitatory flickering open and shut and inhibitory shunting) depending on where they are expressed (neurons or glia) and the stage of development.

## 1. Introduction

The metabolite carbon dioxide (CO_2_) is excreted by breathing. The levels of CO_2_ in the blood normally fluctuate around the physiological level of 40 mmHg, but if the level of blood CO_2_ becomes too high, then the blood will become acidic, which in severe cases can lead to death. CO_2_ levels can be increased in people who suffer from asthma, sleep apnoea or chronic obstructive pulmonary disease (COPD) [1]. In contrast, levels of CO_2_ can be lowered by hyperventilation or when elite athletes drive the levels down during enhanced performance [2]. Changing the level of CO_2_ experimentally (without any compensation) will result in a change in pH. The role of pH-sensitive receptors in the control of breathing is well established, both in the periphery and centrally in the medullary chemosensory areas, such as the retrotrapezoid nucleus and the medullary raphe [3,4,5,6,7,8]. However, there is also evidence that CO_2_ has additional effects on breathing that are independent from pH [9,10].

Connexins are a family of proteins that are embedded in cell membranes and can form large-pored channels. They assemble into large conductance hexameric channels called connexons. Two of these connexons on neighbouring cells can co-assemble to form a gap junction. These channels allow the passage of both ions and other small molecules, such as glucose and lactate. This direct intercellular connection is a well-known form of cell-to-cell communication. If there is just the one connexon that opens into the extracellular space and is not docked to an adjacent cell, this is known as a hemichannel, which has functions that are distinct from those of gap junctions [11,12,13,14,15]. The opening of hemichannels allows the release of adenosine triphosphate (ATP) into the extracellular space to signal via ionotropic P2X and metabotropic P2Y receptors on neurons and glia [12,13]. They can also, through opening, drastically alter both neuronal excitability and whole-cell conductance [16,17].

### CO_2_ Sensing by Connexins

Of the 21 connexin genes in the human genome, a small subfamily has evolved to interact directly with CO_2_ (connexin 26 (Cx26), connexin 30 (Cx30) and connexin 32 (Cx32) [12,18,19,20,21]. Application of CO_2_ in the range from 20 mmHg to 70 mmHg at constant extracellular pH causes dose-dependent opening of the hemichannels of these connexins. Whereas Cx26 and Cx30 have broadly similar sensitivity to CO_2_ with a midpoint around 40 mmHg, Cx32 requires higher doses of CO_2_ for hemichannel opening [12,20,21].

The mechanism of CO_2_ binding to Cx26 has been studied in detail. CO_2_ appears to carbamylate K125. Carbamylation is a spontaneous and labile covalent reaction between a non-protonated primary amine and CO_2_ itself, whereby a covalent bond forms between the C and N atoms. For the primary amine of K125 to be carbamylated, it must be in an environment within the protein that permits access of CO_2_ but that also causes the pK_a_ of the amine side chain to be lowered such that it is predominantly in the non-protonated form. Carbamylation of K125 introduces a negative charge to the side group, and the carbamylated K125 is thought to form a salt bridge with R104 of the neighbouring subunit [20]. K125, R104 and the neighbouring residues comprise a carbamylation motif [20,21].

This model of CO_2_ binding is supported by compelling evidence: for example, if the carbamylation motif is inserted into a CO_2_-insensitive connexin (connexin 31), then CO_2_ sensitivity is induced in these channels [20]. Furthermore, the mutations K125R or R104A within Cx26 abolish CO_2_ sensitivity [20]. The mutations K125E or R104E (to mimic the effect of carbamylation of K125) create a constitutively open hemichannel [20]. If K125 is mutated to C125, then the hemichannel becomes sensitive to NO or NO_2_ instead of CO_2_, and the double mutation K125C and R104C creates a redox-sensitive hemichannel [22]. Finally, there are now cryoEM structures of hemichannels that support the carbamylation of K125 at different levels of PCO_2_ [23].

Although the CO_2_-dependent modulation of Cx30 and Cx32 has not been studied in the same detail as Cx26, these connexins have a very similar carbamylation motif and they are assumed to share the same mechanisms of CO_2_ binding and channel gating. Recently, CO_2_ has been found to close Cx26 gap junctions. This closing action of CO_2_ on the gap junctions shows a similar dependence on K125 and R104 to the hemichannel opening [24]. Interestingly, gap junctions of Cx32 are insensitive to CO_2_, indicating that the actions of CO_2_ on the connexins have considerable complexity.

Since the physiological midpoint for Cx26 hemichannel opening is ~40 mmHg CO_2_, there will be a proportion of Cx26 hemichannels that are open at rest. Raising the level of CO_2_ above this midpoint results in the opening of more Cx26 hemichannels (increasing whole-cell conductance). In turn, decreasing the level of CO_2_ below this midpoint will result in the closing of more Cx26 hemichannels (decreasing conductance) [12].

## 2. Physiological Consequences of Cx26-Mediated CO_2_ Sensitivity

### 2.1. Cx26 in Glial Cells

CO_2_-sensitive connexins are widely expressed across the brain in astrocytes (Cx26 and Cx30) or oligodendrocytes (Cx32) [25,26] rather than neurons. While the function of the CO_2_-sensitivity of Cx30 and Cx32 remains unexplored, Cx26 is an important CO_2_ chemosensor involved in the control of breathing. CO_2_-dependent ATP release is an important mechanism that mediates adaptive changes in ventilation during hypercapnia [3,27]. This ATP release occurs at least in part through Cx26 hemichannels, which are expressed in glial cells at the surface of the medulla oblongata [19]. 

The functional effects of this CO_2_-sensitivity have been studied via an innovative method that used two mutations (K125R and R104A) in the Cx26 gene to create a dominant negative subunit (dnCx26, Figure 1a). These mutations ensure that the dnCx26 subunit can neither bind CO_2_ nor interact with a neighbouring subunit that has bound CO_2_. It has been shown by FRET to co-assemble with wild-type Cx26 and result in heteromeric hemichannels that are unable to change conformation and open in response to CO_2_ [26]. Therefore, it provides a method to remove CO_2_ sensitivity from Cx26 without deleting the gene and, thus, other potential functions of Cx26 (Figure 1a).

When the dnCx26 subunit was expressed selectively in glial cells in a specific area of the medulla, the caudal parapyramidal area (cPPy), it substantially reduced the adaptive ventilatory responses to hypercapnia [26] (Figure 1b). Interestingly, although Cx26 is expressed in glial cells more rostrally, expression of dnCx26 in these cells had no effect on the CO_2_ sensitivity of breathing. Thus, CO_2_ sensing via Cx26 occurs only in a very restricted population of glial cells, located in the cPPy. These cells have superficial cell bodies, and long processes that project rostrally and medially and could enable signal transmission to neuronal populations such as the raphe pallidus and raphe obscurus [26] (Figure 1c). In fact, considering the relative proportions of the ventilatory response mediated by the peripheral chemosensors in the carotid and the central chemosensors [28], CO_2_ binding to Cx26 in the glial cells of this small, circumscribed area contributed nearly half of the adaptive ventilatory response to hypercapnia that is mediated by central chemosensors [26,28]. This result directly links the carbamylation of Cx26 to physiological action (in this case the control of breathing).

### 2.2. Cx26 and Cx32 Are Expressed in Developing Cerebral Cortical Neurons

In the mammalian foetus, the subplate, which is also called the subplate zone, together with the marginal zone and the cortical plate, represent the rudimentary beginnings of the cerebral cortex (Figure 2). Subplate neurons (SP neurons) are among the first neurons generated in the cerebral cortex. Although they disappear during postnatal development, they play an important role in establishing the correct connectivity pattern and are key to the functional maturation of the cerebral cortex. They are also some of the first neurons to receive inputs from thalamic axons. The cortical plate (CP) is the final plate formed in corticogenesis and includes cortical layers 2 to 6. The cortical plate is formed above the subplate (Figure 2).

There is substantial evidence for the expression of CO_2_-sensitive connexins in developing cortical neurons in both rodents and humans. In the rat, Cx26 (the connexin most sensitive to CO_2_) expression is highest prenatally and during the first 3 weeks of life and then expression levels diminish [29]. The Cx26 is localised in the soma and dendrites of neurons, co-labelling with the neuron specific cytoskeleton protein MAP-2 [29]. In contrast, Cx32 (less sensitive to CO_2_, [12]) is only expressed postnatally from day 7 in the rat [29]. In the mouse cortex, at postnatal day 2, there is labelling for Cx26 in both CP and SP zones, but this appears to be non-neuronal. At postnatal day 10 in the mouse, Cx26-labelling is co-localised with the neuronal nuclear marker NeuN in both CP and SP [30]. In the human foetal cortex, Cx26 mRNA is present at gestational weeks (gw) 17 and 19 [16]. At around gw 17, the foetus is ~12 cm long and weighs about 150 g. Cx32 labelling is observed in individual cells of the SP zone during gw 20–22 [16]. In humans, both Cx26 and Cx32 expression can be observed in neuronal processes and cell bodies. 

SP neurons in the developing cerebral cortex of rodents and humans exhibit spontaneous plateau depolarisations. These depolarisations reach threshold, and thereforelead to the firing of action potentials [16,30,31]. It is proposed that this early spontaneous activity is important in driving processes that increase gene expression and to maintain the synaptic contacts between neurons. To investigate the mechanism for the production of this spontaneous activity, a number of experiments have been carried out. The spontaneous activity could be produced by neural network activity driving the release of neurotransmitters which produce depolarisation, it could be produced by gliotransmitter release from glial cells following calcium waves or it could be produced via gap junctions between neurons. A cocktail of synaptic blockers (antagonists for glycine, AMPA, NMDA and GABA-A receptors) did not change the number of spontaneous depolarising events or their durations in the mouse brain [30]. The depolarisations were also insensitive to pannexin and voltage-gated calcium channel blockers. However, the spontaneous events could be blocked by non-selective inhibitors of gap junctions and hemichannels (such as octanol) and by inhibitors of glial metabolism (DL-fluorocitrate). The specific blocker of hemichannels, lanthanum (La^3+^) also significantly inhibited the spontaneous activity [30], confirming a role for hemichannels. However, blockers of hemichannels could be acting within the network (on neurons and glia) to, for example, inhibit the release of transmitters, such as ATP via hemichannels, or the hemichannels could be present on the SP neurons themselves. To test this, the hemichannel blocker Gd^3+^ was directly introduced into neurons via the patch pipette and this caused a loss of spontaneous activity. This directly shows that Gd^3+^-sensitive channels (which are probably hemichannels) are present on the membrane of SP neurons [30]. However, although the trivalent ions gadolinium (Gd^3+^) and lanthanum (La^3+^) block hemichannels without inhibiting gap junction channels or Panx1 channels, they do inhibit other channels including maxi-anion channels [32] and Ca^2+^ channels [33]. Thus, further experiments are required to fully define their role in the generation of spontaneous depolarisations.

Human foetal SP neurons also display spontaneous depolarisations (similar to those observed in rodents) in the period of 17–23 gestational weeks [16,31]. The action potentials on top of the plateaus are blocked by tetrodotoxin (a voltage gated sodium channel inhibitor) but the plateaus themselves are unaffected. Thus, the plateaus are not produced by action potentials and the action potential-dependent release of neurotransmitters This is confirmed as the plateaus persist in the presence of GABA and glutamate receptor antagonists. However, gap junction and hemichannel inhibitors, i.e., octanol, flufenamic acid and carbenoxolone, significantly reduced the spontaneous plateau potentials. The putative hemichannel antagonist, lanthanum, applied alone was also a potent inhibitor of the spontaneous activity [16,34,35].

The importance of Cx26 in cortical development is illustrated by its removal impairing cortical development [36]. Conditional deletion of *Cx26* in the superficial layer excitatory neurons of the mouse neocortex around birth significantly reduced spontaneous firing activity and the frequency and amplitude of spontaneous network oscillations at postnatal day 5–6. Moreover, *Cx26*-conditional knockout (*CX26*-cKO) neurons had simpler dendritic trees and lower spine density compared with wild-type neurons. However, this knockout will affect both gap junctions and hemichannels, and thus, the precise role of hemichannels in development is not clear.

There have been no experiments testing the effects of changing CO_2_ levels on the spontaneous activity present in these developing cortical neurons. One would predict that changing CO_2_ levels would modulate hemichannel opening (particularly the Cx26 hemichannels), and therefore, they could change spontaneous electrical activity. Thus, CO_2_ levels may play an important role in cortical development. How do the hemichannels produce the spontaneous depolarising plateau potentials? One can speculate that that the brief flickering of hemichannel openings (proposed by [16], see Figure 2) and directly observed by [12] could lead to transient cation influx producing the plateau depolarisations. Longer, more sustained openings would lead to a fall in the input resistance and a reduction in excitability (as seen in adult neurons, outlined below).

### 2.3. Cx26 in Adult Neurons

We have recently provided the first evidence for functional CO_2_-sensitive connexin hemichannel expression in adult neurons [17]. Dopaminergic neuron (DN) coupling via gap junctions was first described by [37]. A subsequent study [38] highlighted the presence of functional gap junctions between pairs of neighbouring dopaminergic neurons in the substantia nigra, a region involved in the regulation of movement and the sleep-wake cycle [39]. In a subsequent study they examined which connexins are expressed by these dopaminergic neurons. At postnatal day 7–10, connexins 26, 30 36, 23 and 43 are expressed (mRNA, obtained from single cell RT-PCR). This is followed by a developmental shift, and by postnatal days 17–21, there is only expression of Cx31.1, 36 and 43 [40]. This shift is of considerable interest, given that connexins 26 and 30 are directly sensitive to changes in CO_2_ levels, and thus, the neurons may be sensitive to CO_2_ during development. In our recent study [17], we used a combination of electrophysiology, dye loading and immunohistochemistry to evaluate whether this connexin expression profile resulted in neuronal sensitivity to CO_2_ [41].

When the level of CO_2_ was increased (from 35 mmHg to 55 mmHg CO_2_ at constant extracellular pH), we observed a marked increase in whole-cell conductance (indicating the opening of a membrane channel) of substantia nigra (SN) dopaminergic neurons from postnatal day 7–10 mice. The opening of membrane channels reduces neuronal resistance, leading to a loss of current (Figure 3a,b). Thus, for a given current, there will be a reduction in the voltage response, and thus, a fall in excitability. In the same age of mice, the opposite effect (decrease in whole-cell conductance of SN dopaminergic neurons) was observed when CO_2_ was reduced (from 35 mmHg CO_2_ to 20 mmHg CO_2_), increasing the excitability. These effects of changing CO_2_ were not observed in older mice (postnatal day 17–20, Figure 3c,d). These observations are consistent with the developmental connexin expression profile outlined in Vandecasteele’s independent study [40] and also consistent with the involvement of Cx26 as the mediator of this conductance change. This was further supported by demonstrating CO_2_-dependent loading of a membrane impermeant dye into these cells (Figure 3e) and the block of the increase in conductance by the hemichannel inhibitor carbenoxolone and the immunohistochemical localisation of Cx26 in SN dopaminergic neurons (Figure 3f).

The ventral tegmental area (VTA) is highly heterogeneous and a core region of dopaminergic signalling in the brain. It has subgroups of neurons that can singularly release dopamine, glutamate or GABA or co-release a combination, allowing the VTA to function flexibly [42,43,44,45,46,47]. We have also discovered that a subpopulation of VTA GABAergic neurons is directly sensitive to levels of CO_2_. Similar to SN neurons, increasing CO_2_ increased whole-cell conductance (reducing excitability), whereas reducing CO_2_ decreased whole-cell conductance (increasing excitability). These neurons could also be dye loaded with a membrane-impermeant dye when the channels were opened. We have further demonstrated that the sensitivity of neurons in VTA neurons are mediated via Cx26 membrane hemichannels. This could have important implications in several SN- and VTA-mediated behaviours, including movement, reward, sleep/wake and arousal [42,48,49,50,51,52,53]. 

One can hypothesise that an increase in CO_2_ (which occurs in conditions such as sleep apnoea) would inhibit the VTA GABA neurons leading to arousal (waking up). We know that activating the VTA GABA neurons leads to sleep [53], and thus, their inhibition would be predicted to produce arousal.

### 2.4. Developmental Shift in Cx26 Expression Mediates a Change in Signalling in the Substantia Nigra

As mentioned above, using dye loading and immunohistochemistry, we observed that as connexin 26 expression was lost from dopaminergic neurons in the substantia nigra (after postnatal day 10). However, using a marker for glial cells (GFAP), we found that the expression shifted to neighbouring glial cells [17] (Figure 3g). Given the responses to neuronal excitability by raised CO_2_ (directly inhibitory if Cx26 expressed in neurons and indirectly excitatory if Cx26 expressed in glia), this suggests a shift in Cx26-mediated signalling within the substantia nigra occurs during development. This highlights a key difference from the VTA, where the Cx26 expression remains in the GABAergic neurons throughout development. It can be speculated that the differences in Cx26 expression in the SN vs VTA may possibly contribute the enhanced vulnerability of SN DNs in diseases such as Parkinson’s disease. 

## 3. Conclusions

CO_2_-sensitive connexin hemichannels have at least two modes of operation: (1) they allow the release of mediators such as ATP from glial cells, which can then diffuse in the extracellular space to excite neighbouring neurons and glia, and (2) they can directly increase the conductance of neurons, reducing their excitability. Both of these effects occur in response to increases in CO_2_. Since the mid-point of hemichannel opening is related to the physiological levels, some hemichannels will be open at rest, and thus, small increases and decreases in CO_2_ can modulate signalling minute to minute. There is also a possible third mode of signalling: there is strong evidence that CO_2_-sensitive hemichannels are expressed by developing cortical neurons. These hemichannels appear to contribute to the generation of spontaneous activity that is required for cortical development. It is plausible that changes in CO_2_ levels will modulate this activity and could, therefore, play a role in cortical development. It is possible that the transient opening of the hemichannels leads to depolarisations via cation influx, although this has not been directly tested. Thus, CO_2_-sensitive hemichannels provide a surprisingly complex form of signalling (Figure 4) with mechanisms that include the release of mediators, neuronal shunting and neuronal depolarisation depending on hemichannel expression pattern and stage of development.

## Figures and Tables

**Figure 1 ijms-22-07254-f001:**
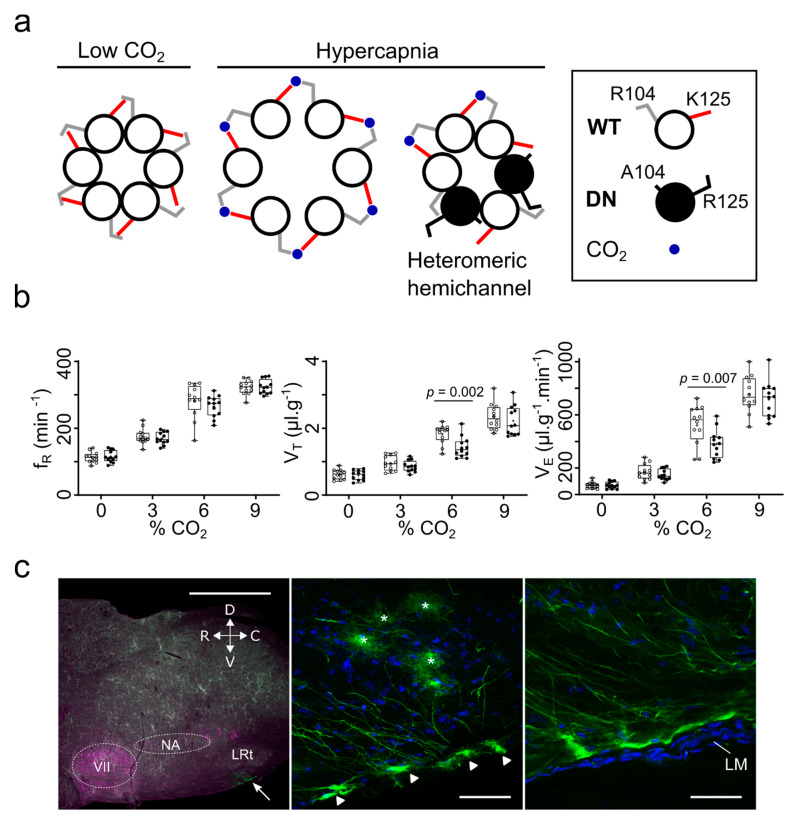
The contribution of Cx26 to the chemosensory control of breathing. (**a**) The concept of the dominant negative Cx26 subunit (Cx26^DN^) which carries the mutations R104A and K125R. Coassembly of Cx26^WT^ and Cx26^DN^ into heteromeric hemichannels will be insensitive to CO_2_ because insufficient carbamate bridge formation will occur to induce channel opening. (**b**) Expression of Cx26^WT^ (open circles) and Cx26^DN^ (filled circles) under control of GFAP promoter in the caudal parapyramidal area reduces the response to hypercapnia. Both the hypercapnia-induced increases in tidal volume (V_T_) and minute ventilation (V_E_) are reduced by Cx26^DN^ expression. Cx26^DN^ expression has no effect on the respiratory frequency (f_R_). (**c**) Location of transduced cells. Left image: cholinergic cells stained with anti-ChAT; the white arrow points to superficial transduced glial cells ventral to the lateral reticular nucleus (LRt); scale bar 1 mm. The centre and right images show these cells at higher power (scale bars 50 µm). Asterisks indicate astrocytes, triangles the superficial somata of the chemosensory glial cells. VII—facial nucleus; NA—nucleus ambiguous; LM—leptomeningeal layer. Adapted from [26].

**Figure 2 ijms-22-07254-f002:**
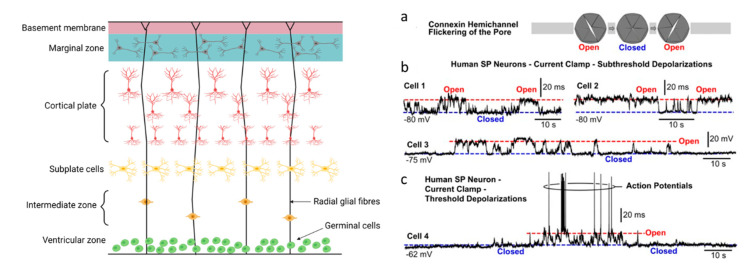
Hemichannels in subplate neurons: flickering of the connexin pore. Left panel, diagram of the developing cortex showing position of cortical and subplate neurons. Created with Biorender. Right panel, Spontaneous flickering of Cx pores. (**a**) Schematic drawing of a Cx hemichannel. Neuronal depolarisations occurs when the connexin pore is in an open state. (**b**) Current clamp recordings of spontaneous depolarisations in human SP neurons. Cells 1–3 are hyperpolarised with negative holding current, so that AP firing threshold is not reached. (**c**) Current clamp recordings of spontaneous depolarisations in human SP neurons. Cell 4 is closer to the AP firing threshold. Taken from [16].

**Figure 3 ijms-22-07254-f003:**
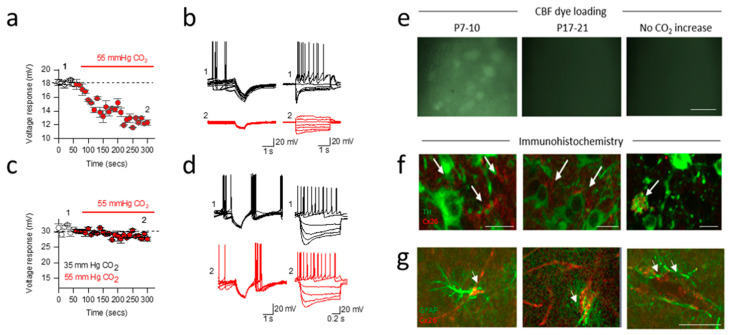
CO_2_ sensitivity of dopaminergic neurons in the substantia nigra alters across development. (**a**) Changes in voltge responses to hyperpolarising current steps for a neuron from a P7-10 mouse when CO_2_ was increased from 35 to 55 mmHg (red). (**b**) Associated voltage traces (40 superimposed traces) in response to step currents (from −200 pA to 50pA) at the indicated time points from a. (**c**) There was no significant change in the voltage response to hyperpolarising current steps in a dopaminergic neuron from a P17-21 mouse when CO_2_ was changed from 35 to 55 mmHg. (**d**) Associated voltage traces (40 superimposed traces) and voltage responses to step currents at indicated time points from c. (**e**) Carboxyfluorescein (CBF) dye loading following hypercapnia (55 mmHg CO_2_) in P7-10 mouse slices. No dye loading occurred if CO_2_ was not changed or if CO_2_ was increased in P17-21 slices, scale bar = 50 µm. (**f**) Immunofluorescent staining of P7-10 SN for Cx26 (red, arrows) in TH+ neurons (green) in single optical planes, scale bar = 20 µm. (**g**) Immunofluorescent staining of P17-21 SN for Cx26 (red, arrows) in glial fibrillary acidic protein (GFAP; green) in single optical planes, scale bar = 50 µm. Adapted from [17].

**Figure 4 ijms-22-07254-f004:**
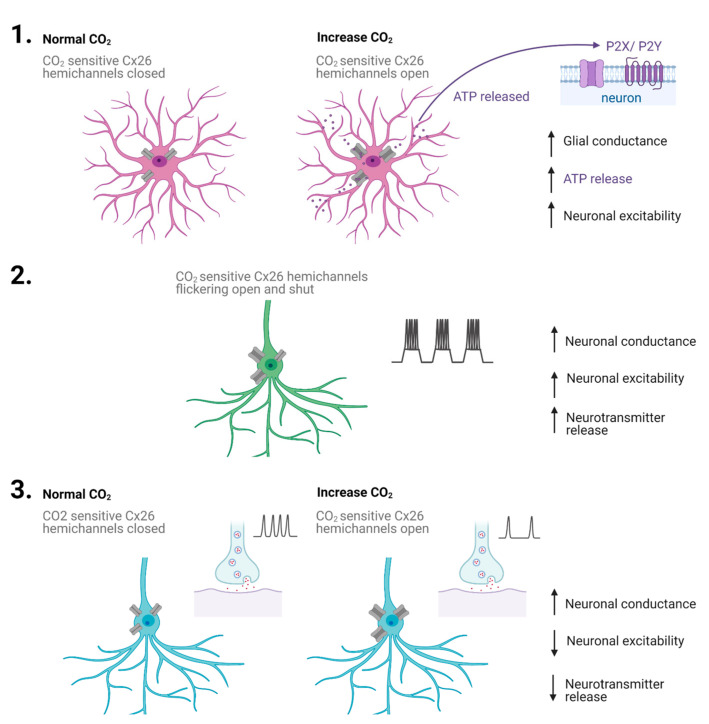
We propose that connexin 26 (Cx26) has three signalling modes: (1) In glia, when CO_2_ is increased, Cx26 hemichannels will open, resulting in the release of ATP, which can then bind and signal through P2X ionotropic and P2Y metabotropic receptors on neighbouring neurons and glial cells, increasing the excitability. (2) In developing cortical neurons, the flickering of hemichannels from closed to open may lead the depolarising plateau potentials, which are important for development. This flickering may occur when CO_2_ is at basal levels. (3) In young substantia nigra neurons and adult VTA neurons, increases in CO_2_ levels lead to a stable increase in neuronal conductance and a decrease in neuronal excitability, which will result in a reduction of neurotransmitter release.

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
