# Peer review of "CO2-Sensitive Connexin Hemichannels in Neurons and Glia: Three Different Modes of Signalling?"

_ijms, 2021, doi:10.3390/ijms22147254_

Round 1
Reviewer 1 Report
The review is well written and the citations seem to be appropriated. I would add a few more basic references, for example, for the use of classical Connexin's inhibitors as Lanthanide or Gadolinium.
On line 166, there is a parenthesis sign out of place.
Author Response
The review is well written and the citations seem to be appropriated.
I would add a few more basic references, for example, for the use of classical Connexin's inhibitors as Lanthanide or Gadolinium.
We have added some additional references on the block of hemi-channels by these cations.
On line 166, there is a parenthesis sign out of place. This has been corrected
Reviewer 2 Report
As a continuation of previous study, Authors aim to classify different modes of connexin (Cx) channel signalling from known structural, cellular and developmental data along with explaining the rather complex CO2 sensitivity of connexin hemichannel (Cx HC, connexon) and gap junction open/closed functions. It is hypothesized that mutation of R104 and K125 reduces/impairs CO2 sensitivity of the subset of Cxs (Cx26, Cx30 and Cx32) claimed to be relevant in this respect. The reviewer acknowledges the importance of finding structural correlates of CO2 sensitivity, however, calls the specificity of alteration of R104 and K125 residues within the second trans-membrane helix of Cx26 into question. Taking the astroglial Cx43 as an example, we can observe K144 in Cx43 corresponding K125 in Cx26, and K105 in Cx43 corresponding R104 in Cx26. These data conclude to the suggestion that a single substitution of K105 for R104 shall convey CO2 sensitivity. Authors are invited to give an explanation for this singularity. The Abstract does not describe the proposed three types of Cx channel signalling. Editing of Figures may be improved.
Author Response
This is the response to the first point:
- Cx43 is not CO2 sensitive (Huckstepp et al 2010)
- An alignment tool will pick out K105 and K144 of Cx43 as conserved residues equivalent to R104 and K125 of Cx26. However what really matters is the 3 dimensional orientation of the residues with respect to each other. Cx43 has a much larger intracellular loop than Cx26 (or Cx30 and Cx32) and the equivalence of these residues is only achieved by inserting gaps (equivalent to 14 residues) into the sequences of Cx26 etc. It seems unlikely therefore that the necessary 3D orientation between K144 and K105 of the neighbouring subunit is achieved but this can only be established with a high resolution structure for Cx43. A further, more subtle, point is that the chemical environment of K125 is critical for it to be carbamylated (it needs to be in the deprotonated form implying that the pKa for the amine of the side chain is modified from its value in free aqueous solution). This chemical environment is likely to be influenced by the intracellular loop -given the big differences between the intracellular loops of Cx43 and Cx26 it seems unlikely that this occurs, but again only experimental evidence would show this definitively.
- Similarly to this, Cx31 which is not CO2-sensitive has an equivalent K104 and K residues at positions 122 and 123. Presumably, as these are not appropriately oriented (and may not be carbamylated) this connexin is not sensitive to CO2 (Meigh et al 2013).
- What is not open to speculation is that we have experimentally documented the effect of mutating R104 and K125 in multiple ways on CO2 sensitivity of Cx26 (Meigh et al 2013) including changing these residues to Cys to impart NO and redox sensitivity (Meigh et al 2015). This includes gain of function by transplanting the relevant residues into the non-CO2 sensitive Cx31 (including extra residues to ensure the correct orientation of K125 in the mutated Cx31, Meigh et al 2013). Note also as mentioned in point 3 Cx31 has K104, showing that either K104 or R104 can support the carbamate bridging mechanism proposed. Thus the lack of CO2 sensitivity of Cx43 is not due to K at this position. I note also that the mutations R104E or K125E give constitutively open Cx26 hemichannels (in effect engineering CO2 into the molecule via the Glu side chain, Meigh et al 2013).
- Finally we have demonstrated the carbamylation of K125 (Brotherton et al 2020).
- Detailed evolutionary analysis of Cx26, Cx32 and homologous proteins from fish, amphibians, reptiles shows the importance of the carbamylation motif (Dospinescu et al 2019).
The Abstract does not describe the proposed three types of Cx channel signalling.
We have modified the abstract so that the three types of Cx channel signalling are described.
Editing of Figures may be improved.
We have changed the labelling of the figures and legends so they are consistent.
Round 2
Reviewer 2 Report
The Author's response is fine, the MS is acceptable by now.